# Immunotherapy Discontinuation in Metastatic Melanoma: Lessons from Real-Life Clinical Experience

**DOI:** 10.3390/cancers13123074

**Published:** 2021-06-20

**Authors:** Nethanel Asher, Noa Israeli-Weller, Ronnie Shapira-Frommer, Guy Ben-Betzalel, Jacob Schachter, Tomer Meirson, Gal Markel

**Affiliations:** 1Ella Lemelbaum Institute for Immuno-Oncology in Sheba Medical Center, Ramat-Gan 526000, Israel; Ronie.Shapira@sheba.health.gov.il (R.S.-F.); Guy.Ben-Betzalel@sheba.gov.il (G.B.-B.); Jacob.Schachter@sheba.health.gov.il (J.S.); Tomer.meirson@biu.ac.il (T.M.); 2Davidoff Cancer Center, Rabin Medical Center, Petach Tikva 49100, Israel; Noais1@clalit.org.il; 3Sackler Faculty of Medicine, Tel Aviv University, Tel-Aviv 6997801, Israel; 4Azrieli Faculty of Medicine, Bar-Ilan University, Zfat 1311502, Israel; 5The Department of Clinical Microbiology and Immunology, Sackler Faculty of Medicine, Tel Aviv University, Tel-Aviv 6997801, Israel

**Keywords:** melanoma, immunotherapy, complete response, treatment discontinuation

## Abstract

**Simple Summary:**

Metastatic melanoma patients derive unprecedented benefit from immunotherapy, and some of them are even considered cured. Currently, there is no consensus on the safety nor on the timing of treatment discontinuation in this population. This is a real-world study on 106 advanced melanoma patients who were treated with immunotherapy for a median of 15.2 months, and who discontinued treatments in the absence of disease progression. We found that after a median follow up of 20.8 m from discontinuation, 32% had progressed. The results of this study reveal the key factors to bear in mind when considering an elective treatment cessation. Namely, patients with non-CR as best response and patients treated in an advanced-line setting should be treated for longer periods, and elective discontinuation should not take place prior to 18 m.

**Abstract:**

Background: Immunotherapy has revolutionized outcomes for melanoma patients, by significantly prolonging survival and probably even curing a fraction of metastatic patients. In daily practice, treatment for responding patients is often discontinued due to treatment-limiting toxicity, or electively, following a major tumor response. To date, the criteria for a safe stop and the optimal duration of treatment remain unclear. Patients and methods: This is a real-world single-site cohort of 106 advanced melanoma patients who were treated with immunotherapy and who discontinued treatments in the absence of disease progression. Here, we describe their long-term outcomes, and analyze the differential characteristics between patients who ultimately experienced progression and those who remained in unmaintained durable response. Results: Patients were treated with anti-PD-1 monotherapy (81%) or in combination with ipilimumab (19%) for a median of 15.2 m (range, 0.7–42.3 m). Upon discontinuation, 75.5% had achieved a complete response (CR). After a median follow-up of 20.8 m (range, 6–58) from discontinuation, 32% experienced disease progression. Median time to progression was 8.5 m (range, 1.5–37). Response to re-induction with anti-PD-1 was observed in 47%. On multivariate analysis, achieving a non-CR response, immunotherapy given in advanced line, and shorter treatment duration were significantly associated with lesser progression-free survival. Conclusions: This is one of the few reports on real-world melanoma patients who discontinued immunotherapy while responding to treatment. This study reveals the key factors to bear in mind when considering an elective treatment cessation. Specifically, patients with non-CR as best response and patients treated in an advanced-line setting should be treated for longer periods, and elective discontinuation should not take place prior to 18 m.

## 1. Introduction

Immune checkpoint inhibition (ICI) using monoclonal antibodies against the cytotoxic T lymphocyte antigen-4 (CTLA-4), and especially programmed death 1 receptor (PD-1), has revolutionized the treatment of advanced melanoma, dramatically improving overall survival (OS) and progression-free survival (PFS) [1,2,3] compared with standard treatments. Combined ICI yielded an even superior response rate and PFS, and probably OS, compared with each agent alone [4].

The optimal duration of anti-PD-1 therapy in responding patients with metastatic melanoma has not yet been determined. In most clinical trials, treatment continued until disease progression or treatment-limiting toxicity, or for a limited duration of two years per protocol [3,4]. In daily practice, treatment is often discontinued due to treatment-limiting toxicity, or electively, several months to years following a major tumor response [5]. Understanding the long-term outcomes after treatment discontinuation is of great importance for patient–physician decisions. Data on responding patients who discontinue therapy is scarce and widely variable [6,7,8,9,10,11], although evidence for durable responses is consistently accumulating. This questions the need for prolonged treatment in responding patients [3,12,13,14]. For example, follow-up data from Keynote-006 demonstrated that patients with complete response (CR) who completed two years of therapy had similar 2y PFS to CR-patients who completed at least six months of therapy [3].

Furthermore, interest is growing regarding outcomes of anti-PD-1 re-treatment in patients who discontinued treatment and later experienced disease progression. To date, data on re-treatment is lacking and based only on small cohorts [6,8,15,16,17].

We conducted a retrospective analysis of advanced melanoma patients who discontinued immunotherapy in the absence of disease progression and assessed their characteristics and long-term outcomes [18].

## 2. Patients and Methods

### 2.1. Patients and Study Design

The study population was comprised of patients with unresectable or metastatic cutaneous melanoma who were treated with anti-PD-1 monotherapy or in combination with anti-CTLA-4 at the Ella Lemelbaum Institute for Immuno-oncology, at Sheba Medical Center, Israel, between January 2014 and May 2019. The population included only patients who discontinued treatment in the absence of disease progression, whether due to toxicity or electively following a major tumor response. The data was derived from our melanoma registry—a single-center prospectively updated oncologic registry. We collected demographic, pathologic, and clinical data. Treatment characteristics included regimen, line of treatment, duration, and reason for discontinuation. Tumor responses were defined as CR, partial response (PR), stable disease (SD), or progressive disease (PD) as determined by the radiologic report, documented in the patient′s electronic file. Data on immune-related adverse events (irAEs) was graded according to the Common Terminology Criteria for Adverse Events (CTCAE) v.5.0. Additional data on progression after discontinuation of treatment and re-treatment was collected.

### 2.2. Statistical Analysis

Differences among quantitative variables were evaluated using the unpaired Student′s *t*-test, whereas Pearson′s Chi-square was used to evaluate differences among categorical variables. Associations were assessed with logistic regression. OS and PFS were estimated from initiation of immunotherapy to death, and to progression or death, respectively. We used Kaplan–Meier methods to estimate and visualize survival and Cox proportional hazards regressions to assess association with baseline prognostic factors. The restricted mean survival time (RMST) and RMST difference (RMST-D) were calculated using the R package survRM218. RMST estimates were truncated at the minimum time point of the largest observed survival time between the two groups. We used the Benjamini–Hochberg procedure to adjust for multiple comparisons of hazard ratio (HR) and RMST-D treatment effects. Statistical significance was defined as *p* ≤ 0.05, and all tests were two-sided. All analyses were performed with STATA v.13.0 and the R statistical software v.3.6.3.

### 2.3. Ethics

This single-center, retrospective study of medical records was approved by the Institutional Review Board of the Sheba Medical Center (4387-17-SMC).

## 3. Results

### 3.1. Patient and Treatment Characteristics

We identified 106 patients with advanced cutaneous melanoma who were treated with immunotherapy and discontinued treatment in the absence of disease progression. Baseline demographic data are detailed in Table 1. The median age was 63 years (range 11.4–88.6), and 67 patients (63%) were male. Twenty-seven patients (25%) harbored a BRAF V600 mutation. As expected, the majority of the cohort had an ECOG performance status of 0–1 (94.3%) and the mean number of disease sites was low, at 1.97 (±1.08). Seven patients (6.6%) had brain metastases (M1d). The discontinued treatment was given as an advanced line in 24.5% of the cohort. The majority of the cohort (81%) was treated with monotherapy.

Upon discontinuation, the most frequent best response was CR (75.5%). Median treatment duration was 15.2 months (range, 0.7–42 m) and was similar between CR, PR, and SD patients. Major reasons for treatment discontinuation, as defined by the oncologist, were treatment-limiting toxicity in 60 patients (56.6%), CR in 32 patients (30.2%), and long-term PR in 14 patients (13.21%). Of the 60 patients who discontinued treatment due to toxicity, 42 (70%) had also achieved CR.

The irAEs, severity, onset, and duration are specified in Figure 1 and Figure 2. Thirty-eight patients (35.8%) experienced high grade (grades 3–4) AEs. While most of the irAEs were temporary, 41 (38.6%) patients developed permanent irAEs, some debilitating. Those were vitiligo (*n* = 24, 22.6%), endocrine disorders (*n* = 17, 16%), neuropathy (*n* = 3, 2.8%), chronic renal-failure (*n* = 2, 1.8%), arthralgia (*n* = 1, 0.9%), and chronic immune thrombocytopenic purpura (*n* = 1, 0.9%). Interestingly, 60 patients (56.6%) were exposed to steroidal therapy.

### 3.2. Outcome after Treatment Discontinuation

After a median follow-up of 39.1 months (range 7–74 m) from treatment initiation, and 20.8 months (range 6–58 m) from treatment discontinuation, 34 patients (32%) experienced disease progression. Median time from treatment discontinuation to disease progression was 8.5 months (range 1.5–37 m).

Patients who achieved CR as best response were least prone to progression compared to non-CR patients (23.7% vs. 57.7%, *p* = 0.02). Furthermore, CR patients who progressed had a longer median time from discontinuation to progression compared to patients with PR or SD (12 m, 5.9 m and 6 m, respectively). The median PFS for patients with CR, PR, and SD was not-reached (NR), 36.5 m, and 12.8 m, respectively (Figure 3A). The median OS for patients with CR, PR, and SD was NR, NR, and 24.6 m (Figure 3B).

Six of the 34 patients who progressed (17.6%) died from the disease, and three patients died due to other reasons, with no evidence of disease progression (one patient had glioblastoma multiforme, one patient had chronic lymphocytic leukemia, and one patient myelodysplastic syndrome). The pattern of progression included progression in known disease sites (11 patients, 32.3%) and development of lesions in new sites (23 patients, 67.7%). The new metastatic sites were subcutaneous or nodal in 13 patients, lung in 1 patient, abdominal parenchymal organs in 5 patients, and brain in 4 patients.

### 3.3. Treatment Re-Induction

Of 34 patients (73.5%) who discontinued treatment and later experienced disease progression, a subsequent course of systemic treatment was administered in 25 patients (Figure 4). The remaining patients were treated with local modalities only (*n* = 5), died without subsequent therapy (*n* = 2), or were expected to start a systemic treatment (*n* = 2).

Most of the patients (21 patients, 62%) were re-treated with immunotherapy (19 with anti-PD-1, 1 with combination ipilimumab plus nivolumab, and 1 with ipilimumab alone), two patients were treated with BRAF-MEK inhibitors, and two patients with temozolamide. In patients who received re-induction with anti-PD-1, the overall response rate was 47% and disease control rate (DCR) was 68%. The best response to anti-PD-1 re-induction was CR in five patients, PR in four patients, SD in four patients, PD in two patients, and unconfirmed-PD in two. One patient had not yet been evaluated, and one patient had no follow-up. Responses to other treatments are specified in Figure 4.

There was no significant correlation between time off-treatment and response to the re-induction of anti-PD-1 (*p* = 0.67). The median duration of anti-PD-1 re-treatment was 7 months (range, 0.7–18.8 m), and to date, seven patients are still receiving treatment. Reasons for treatment discontinuation were treatment-limiting toxicity in eight (66%) and disease progression in three patients (25%). Of 13 patients who responded to anti-PD-1 re-treatment, only three patients experienced later progression, and no patient died.

### 3.4. Factors Associated with Outcome

We examined the differences in selected baseline characteristics between patients who progressed (*n* = 34) and patients with no disease progression (*n* = 72) following treatment discontinuation. We also examined the association of these factors with survival outcomes.

#### 3.4.1. Best Response

Patients achieving CR had a significantly lower risk of disease progression compared to patients achieving PR or SD (odds ratio (OR) 0.31; *p =* 0.02 for PR versus CR, 95% CI 0.11–0.83). Compared to patients with CR, patients achieving PR and SD had HR for PFS after treatment cessation of 2.48 (95% CI 1.22–5.05; *p =* 0.012) and 7.18 (95% CI 2.42–21.25; *p* < 0.0001), respectively. Specifically, looking at a subpopulation of patients who experienced treatment-limiting high-grade adverse events (*n =* 60), we noticed that those who did not achieve CR (30%) at treatment discontinuation had a higher risk of progression compared to patients who did achieve CR (OR 5.8, 95% CI 1.70–19.67; *p =* 0.005, Table 1). This finding points out the importance of best response when considering treatment re-challenge in patients experiencing high-grade AEs.

#### 3.4.2. Line of Treatment

Patients who received prior treatments had a significantly higher probability of progression after treatment discontinuation compared to patients who were treated in the first line (OR 2.8, 95% CI 1.12–7.02; *p =* 0.027). However, PFS was not significantly affected by treatment line (HR 1.46, 95% CI 0.74–2.87; *p =* 0.279).

#### 3.4.3. Treatment Duration

Among the 72 non-progressors, the median duration of treatment was 15.8 months (range, 0.7–42.8), whereas patients who progressed had a statistically borderline-significant shorter median treatment duration of 8.9 months (range, 0.7–34.3; *p =* 0.07).

To identify the optimal duration of treatment in patients who achieved CR, we calculated the HR for PFS at 3-month time intervals, representing different cutoffs for treatment duration. The Cox model assumes proportional hazards, namely that the ratio of the hazards is constant over time. Given the non-uniform distribution of the hazard curves, this assumption was not met in all cases, limiting the clinical interpretability of the HR [19,20]. Therefore, we also calculated the RMST, which provides an alternative and intuitive approach for quantifying treatment effect without assuming proportional hazards [19,20]. The RMST-D corresponds to the difference between the areas under the survival curves (RMST); the higher the difference, the greater the RMST-D and estimated clinical benefit. The lowest HR and highest RMST-D with significant values were found between 18 and 24 months (Figure 5). The HR at treatment cutoff of 21 months was 0.18 (95% CI 0.05–0.62; adj. *p =* 0.007), and the RMST-D was 13.1 (95% CI 6.26–19.89; adj. *p =* 0.001). That is, responses of patients with CR treated for more than 21 months lasted 13 months longer. Looking at differences in other cardinal prognostic factors between patients who were treated for less or more than 18 months, we found that both groups had similar proportions of best overall responses, and that, surprisingly, the patients who were treated for less than 18 months had a relatively higher percentage of first-line patients (52 (83%) vs. 28 (65%), respectively; *p* = 0.041). These data further reinforce the significance of a longer treatment duration. Twelve patients were treated for ≥24 months prior to treatment discontinuation. Their three-year PFS was 83.3%.

#### 3.4.4. Adverse Events and Steroid Treatment

The rate of patients who experienced grade 3 or 4 AEs was higher among the progressors compared to patients who did not have disease progression (47% and 30.6%, respectively). In univariable analysis, high-grade AEs (grades 3–4) were significantly associated with progression (*p =* 0.049). This was also reflected in survival outcomes, where HR for progression or death was 2.12 (95% CI 1.1–4.1, *p =* 0.025) in patients experiencing grade 3 or 4 AEs, compared to patients experiencing minor (grade 1 or 2 AE) or no AE (Figure 6A). The type of toxicity or number of systems involved did not impact the risk of disease progression.

Patients who received steroid therapy secondary to irAEs were more likely to have disease progression (OR 2.93, 95% CI 1.20–7.15, *p =* 0.018), and had a significantly poorer PFS (HR 2.85, 95% CI 1.4–5.8, *p =* 0.004, Figure 6B).

The swimmers plot in Figure 7 describes the course on- and off-treatment of all 106 patients, stratified into progression status and exposure to steroid treatments. The median duration of steroid treatment was numerically higher in the progressors group, yet it was not statistically significant (26.5w vs. 14.5w, *p =* 0.256). Neither the duration nor the maximal dosage of steroids were associated with disease progression.

#### 3.4.5. Multivariate Analysis

In a multivariable analysis for PFS, we found that the best response (HR 2.46, 95% CI 1.48–4.07, *p* < 0.001), line of treatment (HR 2.20, 95% CI 1.03–4.70), and treatment duration (HR = 0.98, 95%CI 0.97–0.99) remained significant, whereas high-grade AEs and exposure to steroid therapy were not statistically significant (Table 2).

## 4. Discussion

Our study reports the outcomes of a unique cohort of 106 advanced melanoma patients who discontinued treatment in the absence of disease progression, following a major tumor response. This kind of real-world data is of paramount importance because it brings us one step closer to a better understanding of the clinical factors that are associated with outcomes in this specific population, and therefore should guide the difficult decision of treatment discontinuation.

After a median follow-up of 20.8 m from treatment discontinuation, 32% of our cohort experienced disease progression. As expected, the risk for progression was significantly associated with the best overall tumor response. Patients who achieved CR had a significantly lower risk for progression compared to patients who did not achieve CR (23.7% vs. 57.7%); they also had a longer PFS. This finding is consistent with previous reports [2,3,6,21] and accentuates the importance of best tumor response when considering a permanent discontinuation of treatment.

Previous therapy exposure was also found to be a predictive factor for relapse. In our cohort, 75% of patients were treatment-naïve. These patients had a lower risk of disease progression compared to previously treated patients. This was also shown in recent reports from clinical trials and retrospective cohorts [2,3,6,21].

Since the approval of immune checkpoint inhibitors in melanoma, the optimal duration of treatment has always been an unanswered question, despite its crucial clinical significance. Recent studies published thus far have been inconclusive. Jansen et al. found that among melanoma patients who achieved CR, those who were treated for less than 6 months had a significant higher risk of progression [6]. Conversely, Warner et al. showed no association between treatment duration and progression among patients who achieved CR [8]. Selig et al. also presented similar disease-free survival (DFS) in patients with CR treated with longer treatment courses versus those who stopped therapy prior to seven months [10]. Further, Keynote-006 reported similar PFS estimates for patients with CR who completed six months of treatment and those who completed 2 years [3]. In a recent perspective by Robert et al. [22], it was suggested that immunotherapy may be safely stopped in CR patients if at least 6 months of treatment have been administered after a complete response was confirmed.

Notably, all studies demonstrating no significant benefit for longer treatment duration calculate PFS or DFS from completion of treatment and not from treatment initiation. This leads to a time-zero bias, a form of selection bias with unintended systematic differences between groups at the beginning of the study [23]. In our work, we chose to calculate the PFS from treatment initiation to disease progression, giving weight to time (on- and off-treatment) without progression. We found that within CR patients, a cut-off of 17 to 24 months of treatment provides a significant PFS benefit compared to shorter treatment periods. Furthermore, looking at the whole cohort, HR for PFS was significantly lower for patients treated for more than 18 months, at 0.34 (*p =* 0.003). Therefore, our findings support the approach of treatment discontinuation not prior to 18 months, even in patients who achieved CR. The same analysis was done for PFS calculated from treatment discontinuation. With this method, no significant PFS benefit was demonstrated for any timepoint (Appendix A). We believe that a re-calculation of previous studies, using PFS from the beginning of the treatment rather than from discontinuation, may alter their conclusions, possibly favoring longer treatment durations. In summary, taking the results of our study together with other similar works, no clear-cut conclusion can be drawn as to the optimal treatment duration. This is also due to treatment inconsistencies, inherent selection biases, retrospective nature, small cohorts, and more importantly, variations among patient characteristics. This emphasizes the need for larger-scale real-life studies and possibly randomized controlled clinical trials.

IrAEs have been found to be associated with favorable outcomes in melanoma [24,25]. These toxicities are often treated with corticosteroids. Interestingly, in our cohort, the percentage of patients who experienced high-grade irAEs was higher among those who eventually had disease progression. Accordingly, patients who were treated with steroids were at a higher risk of disease progression, regardless of the maximal dosage or duration of exposure. We believe that exposure to steroids may explain the higher rate of progression in patients with severe irAEs. There is a concern that exposure to steroids may reduce the efficacy of immunotherapy due to their immunosuppressive effects. However, there is inconsistency in the literature regarding the impact of steroidal therapy on treatment outcomes. While several reports have identified an association between a worse outcome and prolonged exposure to high-dose steroids [26] or early exposure to steroids [27], others did not find any negative interactions [24,28]. The impact of exposure to steroids in patients with durable responses for whom treatments were discontinued has not been reported thus far, and merits further investigation.

When deciding on treatment discontinuation, one must consider the efficacy of re-treatment in case of disease progression. Data is lacking and is mostly based on small cohorts. In the last update of the Keynote-006 study, 15 patients were treated with a second course of pembrolizumab, of whom seven (46.6%) achieved a new anti-tumor response, and DCR was 66.6% [3,17]. Furthermore, Jansen et al. reported that of 19 patients who received a subsequent course of anti-PD-1 treatment, six (32%) patients responded [6]. More recently, in a study by Warner et al., only five (15%) of 34 patients responded to anti-PD-1 re-treatment [8]. Other reports included only 3–8 patients [2,9,12,21,29,30]. Overall, 47% (*n =* 9) of our patients responded to re-treatment with anti–PD-1 therapy. Our results suggest that resuming anti PD-1 treatment at time of progression may provide renewed anti-tumor activity.

## 5. Conclusions

We identified several key factors that are associated with outcome and could assist in the process of decision making when considering permanent treatment discontinuation. Specifically, achievement of PR or SD as best response rather than CR, duration of treatment shorter that 18 months and advanced line of treatment may have a negative effect on the PFS of these patients. Exposure to steroids during treatment or after discontinuation may also have an impact and should be used mindfully.

## Figures and Tables

**Figure 1 cancers-13-03074-f001:**
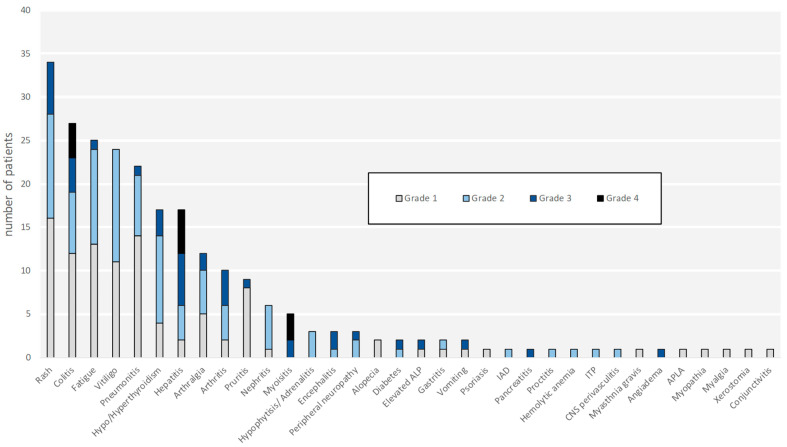
Grade distribution of immune-related adverse events (irAE). The most common irAE observed were cutaneous (29.9%), gastrointestinal (14.2%), rheumatologic (13.4%), endocrine (10.1%), and fatigue (9.7%). Thirty-eight patients experienced high grade (grades 3–4) irAEs. The documented grade 4 irAEs were colitis, hepatitis, and myositis. Abbreviations: irAE—immune-related adverse events; ALP—alkaline phosphatase; IAD—isolated adrenocorticotropic hormone deficiency; ITP—immune thrombocytopenic purpura; CNS—central nervous system; ALPA—antiphospholipid antibody syndrome.

**Figure 2 cancers-13-03074-f002:**
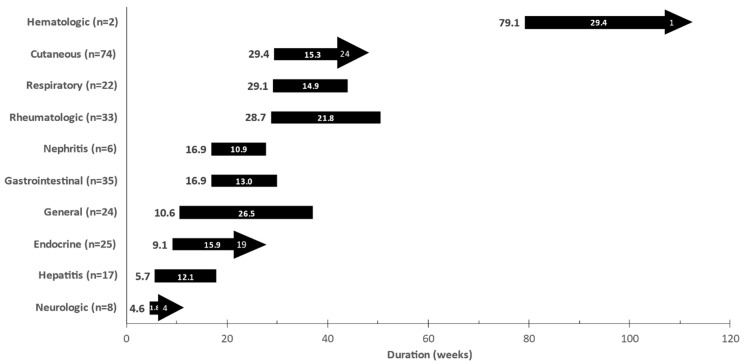
Time to onset and median duration of immune-related adverse events (irAE). The median time to onset of irAE from treatment initiation is shown prior to the bar (e.g., hepatitis 5.7w). The median duration of irAE is represented by the length of the bar (e.g., hepatitis 12.1w). Arrows indicate permanent adverse events. The numeric value within the arrow represents the number of patients with permanent adverse events (e.g., endocrine 19).

**Figure 3 cancers-13-03074-f003:**
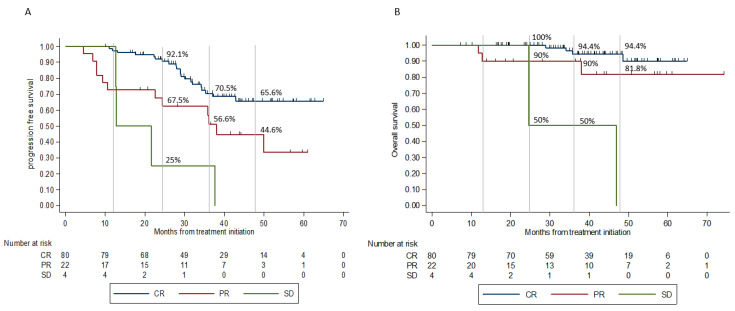
PFS and OS according to best response. (**A**) Progression-free survival according to best response. Patients who achieved CR as best response had median PFS NR, whereas patients who achieved PR and SD as best response had median PFS of 36.5 m and 12.8 m, respectively. Compared to patients with CR, the HR for PFS was 2.48 (95% CI 1.22–5.05; *p =* 0.012) and 7.18 (95% CI 2.42–21.25; *p* < 0.0001) for patients with PR and SD, respectively. (**B**) Overall survival according to best response. The median OS for patients with CR, PR, and SD was NR, NR, and 24.6 m, respectively. Compared to patients with CR, the HR for OS was 2.79 (95% CI 0.62–12.49, *p* = 0.179) and 20.32 (95% CI 3.57–115.5, *p* = 0.001) for patients with PR and SD, respectively. Abbreviations: PFS—progression-free survival; OS—overall survival; NR—not-reached; CR—complete response; PR—partial response; SD—stable disease; PD—progressive disease.

**Figure 4 cancers-13-03074-f004:**
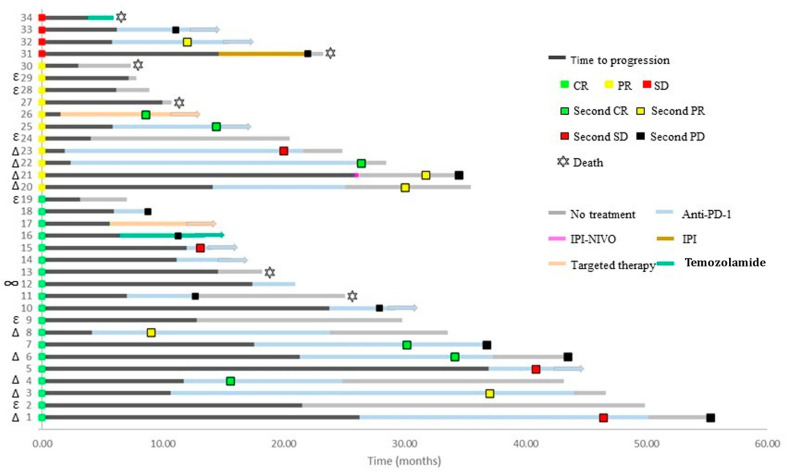
Individual clinical courses for patients with disease progression following treatment discontinuation. Data is presented from treatment discontinuation to last follow-up, second progression, or death. The best response to previous treatment is presented at the onset of the plot, followed by time to progression (time off-treatment), systemic re-treatment, and response. There was no significant correlation between time off-treatment and response to the subsequent treatment; median time off-treatment in patients who responded (PR/CR) to the subsequent treatment was 15 months, whereas for patients who maintained SD or developed PD, the median time to progression was 11.8 months (*p* = 0.95). ^Δ^ In patients 1, 3, 4, 6, 8, and 20–23, re-treatment was discontinued due to treatment-limiting toxicity. ^∞^ Patient 12 was lost to follow-up after re-induction of anti-PD-1. ^ℇ^ Patients who were treated with local therapy: patients 2, 9, and 19 were treated with radiotherapy or surgery to subcutaneous and lung metastasis and are now free of disease; patients 24 and 29 had soft-tissue progression, which was treated with radiotherapy, and are currently planned to start systemic treatment; patient 28 had brain progression treated with SRS and achieved near-CR. Abbreviations: SRS—stereotactic radiosurgery; CR—complete response; PR—partial response; SD—stable disease; PD—progressive disease; PD-1—programmed cell death 1, IPI-NIVO—ipilimumab and nivolumab combination; IPI—ipilimumab.

**Figure 5 cancers-13-03074-f005:**
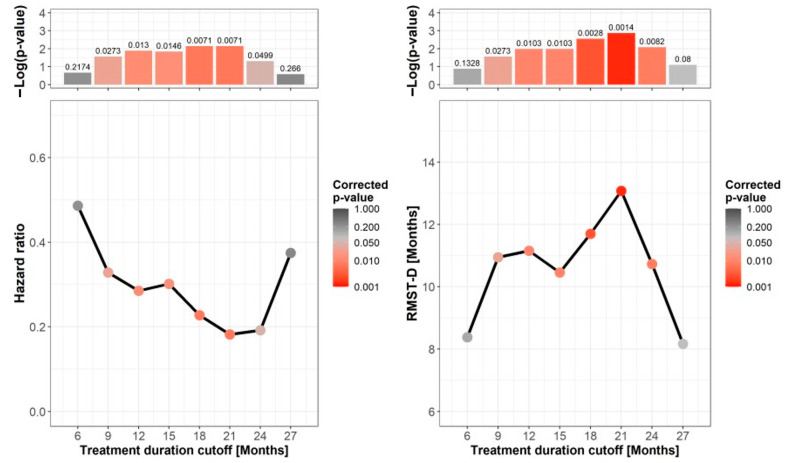
Influence of treatment duration on progression-free survival. Shown are hazard ratio (HR) (**left**) and restricted mean survival time difference (RMST-D) (**right**) for progression-free survival (PFS) of patients with complete response at 3-month time intervals. Each time point represents a duration cutoff that stratifies patients into short or long treatment duration based on the time of treatment discontinuation. The corrected *p*-values are shown in color and presented as columns at the top panels. The lowest HR for PFS and highest RMST-D with significant *p*-values were found between 18 and 24 months of treatment durations.

**Figure 6 cancers-13-03074-f006:**
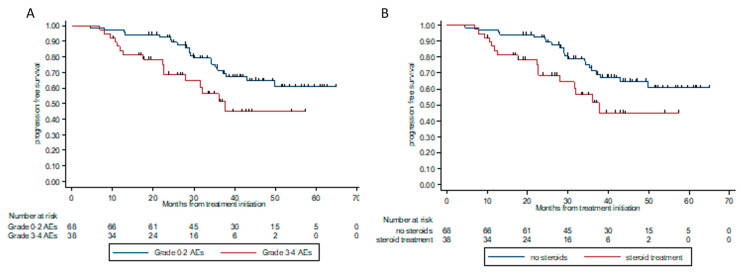
Progression-free survival according to grade of toxicity (**A**) and exposure to steroidal treatment (**B**). (**A**). PFS according to the grade of toxicity. Patients who experienced grade 0–2 AEs had median PFS NR, whereas patients who experienced high grade (grade 3–4) AEs had median PFS of 37 m. HR for PFS was 2.12 (95% CI 1.1–4.1, *p =* 0.025). (**B**). PFS according to exposure to steroidal treatment. Patients who were exposed to steroids higher than 10 mg prednisolone (or equivalent) had median PFS of 37.6 m, whereas patients with no steroid exposure or ≤10 mg prednisolone (or equivalent) had median PFS NR. HR for PFS 2.85 (95% CI 1.4–5.8), *p =* 0.004.

**Figure 7 cancers-13-03074-f007:**
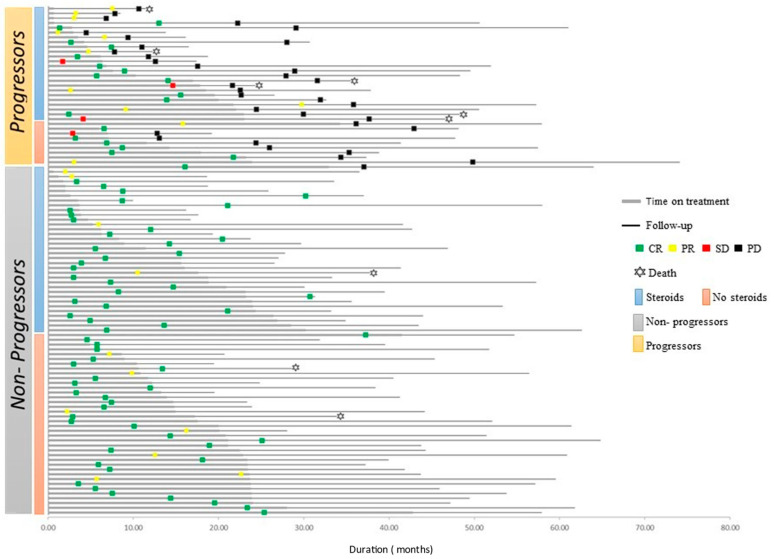
Individual clinical courses of all patients treated with immunotherapy who discontinued treatment in the absence of disease progression. Data is presented from treatment initiation to the last follow-up or death. Patients were sorted according to disease progression and steroid exposure. Patients achieving CR were less likely to have disease progression at treatment discontinuation (24%), compared to patients achieving PR or SD as best response (50% and 100%, respectively; odds ratio (OR), 0.31; *p =* 0.02 for CR versus PR). Median time from treatment discontinuation to progression was 12 m for patients with CR, and 5.9 m and 6 m for patients with PR and SD, respectively. Steroid therapy (prednisolone equivalent dose >10 mg) during treatment was associated with a higher likelihood of disease progression after treatment discontinuation (OR 2.93, 95% CI 1.20–7.15, *p =* 0.018). Abbreviations: CR—complete response; PR—partial response; SD—stable disease; PD—progressive disease; OR—odds ratio.

**Table 1 cancers-13-03074-t001:** Baseline patient characteristics.

Baseline Characteristics	All Patients (*n* = 106)	No Progression (*n* = 72)	Progression (*n* = 34)	*p*-Value
Age (median, range)	63.25 (11.4–88.6)	60.15 (11.4–88.6)	65.5 (27–82.7)	0.085
Male (%)	67 (63.2)	49 (68.1)	18 (52.9)	0.132
BRAF (%)				
V600 mutant	27 (25.5)	19 (26.4)	8 (23.5)	
WT	70 (66.04)	46 (63.9)	24 (70.6)	0.662
unknown	9 (8.5)	7 (9.7)	2 (5.9)	0.669
Metastatic upfront	27 (25.5)	15 (20.8)	12 (35.3)	
Systemic recurrent disease	79 (74.5)	57 (79.2)	22 (64.7)	0.111
Primary melanoma (*n* = 79)				
Breslow (median, range)	2.8 (0.2–18)	2.8 (0.25–18)	2.55 (0.2–17)	0.834
Ulceration (%)	32 (51.6)	23 (58.9)	9 (39.1)	0.131
LDH (%)				
≤UNL	66 (62.3)	47 (65.3)	19 (55.9)	
>UNL	21 (19.8)	15 (20.8)	6 (17.6)	0.985
unknown	19 (17.9)	10 (13.9)	9 (26.5)	0.134
AJCC 8th edition, (%)				
M1a	37 (34.9)	26 (36.1)	11 (32.3)	
M1b	34 (32.1)	25 (34.7)	9 (26.5)	0.76
M1c	28 (26.4)	16 (22.2)	12 (35.3)	0.275
M1d	7 (6.6)	5 (6.9)	2 (5.8)	0.951
Number of disease sites (mean ± sd)	1.97 ± 1.08	1.87 ± 1.09	2.17 ± 1.03	0.181
ECOG PS 0–1 (%)	100 (94.3)	68 (94.4)	32 (94.1)	0.224
**Treatment Characteristics**				
Regimen (%)				
Ipi-Nivo	20 (18.9)	11 (15.2)	9 (26.5)	
Anti-PD-1	86 (81.1)	61 (84.7)	25 (73.5)	0.169
Nivolumab	31 (29.3)	23 (31.9)	8 (23.5)	**-**
pembrolizumab	55 (51.9)	38 (52.8)	17 (50)	**-**
Line of treatment (%)				0.027
1st	80 (75.5)	59 (82)	21 (61.8)	
Advanced *	26 (24.5)	13 (18)	13 (38.2)	
Time on treatment, months (median, range)	15.2 (0.7–42.8)	15.8 (0.7–42.8)	8.9 (0.7–34.3)	0.075
CR	15.5 (0.7–42.8)	16.0 (1.8–42.8)	10.3 (0.72–32.9)	0.1
PR ^†^	12.9 (0.7–34.3)	14.9 (0.69–23.7)	4.8 (0.69–34.3)	0.75
SD ^ℇ^	12.4 (6.4–23)	-	12.4 (6.4–23)	**-**
Best response (%)				
CR	80 (75.5)	61 (84.7)	19 (55.9)	
PR	22 (20.7)	11 (15.3)	11 (32.3)	0.02
SD	4 (3.8)	0	4 (11.8)	**-**
Patients with treatment limiting toxicity, *n* = 60 (%)				
CR	42 (70)	29 (85.3)	13 (50)	
Non-CR	18 (30)	5 (14.7)	13 (50)	0.005
Onset of 1st irAE, weeks (median, range), *n =* 99	7.6 (0.14–104)	8.14 (0.14–104)	5.86 (0.14–68.14)	0.206
irAEs G3–4 (%)	38 (35.8)	22 (30.6)	16 (47)	0.049
Exposure to steroids (>10 mg) (%)	60 (56.6)	35 (48.6)	25 (73.5)	0.018
Duration of steroid exposure (median, range)	22.3 (1.5–230)	14.5 (1.5–143.7)	26.5 (3–230)	0.256
prednisolone equivalent-dose				0.726
>2 mg/kg	4 (6.7)	2 (5.7)	2 (8)	
≤2 mg/kg	56 (93.3)	33 (94.3)	23 (92)	

WT—wild type; UNL—upper normal limit; LDH—lactate dehydrogenase; UNL—upper normal limit; AJCC—American Joint Committee on Cancer; ECOG PS—Eastern Cooperative Oncology Group Performance Status; Ipi-Nivo—Ipilimumab and Nivolumab combination; CR—complete response; PR—partial response; SD—stable disease; irAEs—immune-related adverse events. * Previous lines of treatment were: ipilimumab (*n* = 15), targeted therapy (*n* = 6), pembrolizumab (*n* = 3), ipilimumab and nivolumab (*n* = 2); ^†^ PR vs. Cr *p* = 0.264, ^ℇ^ SD vs. CR *p* = 0.689.

**Table 2 cancers-13-03074-t002:** Multivariable analysis for progression-free survival.

Variable	Hazard Ratio	*p*-Value	95% Confidence Interval
Best tumor response	2.46	<0.001	1.48–4.07
Line of treatment	2.20	0.042	1.03–4.70
Treatment duration	0.98	<0.001	0.97–0.99
High-grade adverse events	0.85	0.702	0.37–1.95
Exposure to steroids	2.16	0.085	0.90–5.19

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
