# Peer review of "Immunotherapy Discontinuation in Metastatic Melanoma: Lessons from Real-Life Clinical Experience"

_cancers, 2021, doi:10.3390/cancers13123074_

Round 1

Reviewer 1 Report

This study aimed to identify factors that may influence relapse of advanced melanoma patients who discontinued immunotherapy in the absence of disease progression. The authors conducted a retrospective analysis and assessed patients’ characteristics and long-term outcomes. The results showed that (1) achievement of partial response or stable disease as best response rather than complete response at time of therapy cessation, (2) duration of treatment shorter that 18 months, (3) advanced line of treatment and (4) exposure to steroids during treatment or after discontinuation were all associated with poorer outcome. Conclusively, a minimum treatment duration of 18 months and caution with exposure to steroids during therapy were identified as key consideration that could assist in the process of decision making when considering permanent treatment discontinuation.

The topic is of strong current interest, not only for the field of melanoma but also of other cancers that are treated with immunotherapy.

Association of better immunotherapy outcome with treatment naïve patients and achievement of complete response at the time of therapy cessation is not a novel finding, as acknowledge by the authors.

However, a cut-off of 18 to 24 months of treatment was found to provide a significant progression free survival compared to shorter treatment periods. These results were obtained calculating progression free survival from treatment initiation instead of treatment completion, the latter methodology introducing a time-zero bias. The authors speculate that the contrasting result of no significant benefit for longer treatment duration found by other studies may be due to this difference in methodologies.

My only concern is in regard to the second sentence of the following paragraph:

“Furthermore, within the sub-population of patients who discontinued treatment due to toxicity (60 patients), and did not resume treatment after AE resolution, 70% had achieved CR as best response. Those patients had a higher risk of progression compared to patients who did not achieve CR (OR 5.8, 95% CI 1.70-19.67; p=0.005, 218 (Table 1).”

This reads as if the patients that have achieved CR are at higher risk of progression compared to the patients that did not achieve CR. However, this does not seem to be the case as out of 42 patients that achieved CR, 13 progressed (31%) whereas out of the 18 patients who did not achieve CR, 13 progressed (72%) (Table 1).

Could the authors clarify this?

Furthermore, could the authors assess whether calculation of progression/disease free survival from treatment initiation rather than treatment completion in those studies demonstrating no significant benefit for longer treatment duration would change these results (i.e. longer treatment does actually benefit outcome after therapy cessation)?

Author Response

A point-by-point response to reviewer 1 comments:

Point 1: My only concern is in regard to the second sentence of the following paragraph: “Furthermore, within the sub-population of patients who discontinued treatment due to toxicity (60 patients), and did not resume treatment after AE resolution, 70% had achieved CR as best response. Those patients had a higher risk of progression compared to patients who did not achieve CR (OR 5.8, 95% CI 1.70-19.67; p=0.005, 218 (Table 1).” This reads as if the patients that have achieved CR are at higher risk of progression compared to the patients that did not achieve CR. However, this does not seem to be the case as out of 42 patients that achieved CR, 13 progressed (31%) whereas out of the 18 patients who did not achieve CR, 13 progressed (72%) (Table 1). Could the authors clarify this?

Response 1: The reviewer is absolutely right. We've changed the sentence to a more comprehensive one: "Specifically, looking at a subpopulation of patients who experienced treatment-limiting high-grade adverse events (n=60), we noticed that those who did not achieve CR (30%) at treatment discontinuation had a higher risk of progression compared to patients who did achieve CR (OR 5.8, 95% CI 1.70-19.67; p=0.005, Table 1). This finding points out the importance of best response when considering treatment re-challenge in patients experiencing high-grade AEs". See lines 218-225.

Point 2: Furthermore, could the authors assess whether calculation of progression/disease free survival from treatment initiation rather than treatment completion in those studies demonstrating no significant benefit for longer treatment duration would change these results (i.e. longer treatment does actually benefit outcome after therapy cessation)?

Response 2: We believe that calculating the PFS from the point of treatment discontinuation as opposed to treatment initiation might cause an un-intended lead-time bias, thus neglecting the effect that the time on-treatment may have on the patient's probability of disease progression. We believe that a more accurate PFS estimation is achieved by adding the time on-treatment to the time off-treatment. This is explained in the discussion, lines 333-338. We added a paragraph in the discussion, lines 344-348, stating that no significant PFS benefit was observed at any timepoint of treatment duration, when calculating the PFS from treatment discontinuation. We added the relative figure as supplementary data (lines 471-480).

Reviewer 2 Report

Asher an coauthors retrospectively investigated 106 patients with advanced melanoma who discontinued ICI in the absence of PD (mainly due to response or adverse events). After a median follow up of 20mo, 34 patients experienced PD, 21 patients were retreated with ICI, 9 patients responded again. PD was associated with non-CR due to ICI, higher line ICI and duration of ICI <18-24 months.

This are overall interesting and relevant observations, and the authors did a very diligent analysis of their patients.

I do have some issues to revise the manuscript:

1) I am surprised that there are no differences with regard to Nivo+Ipi versus PD1 monotherapy as primary ICI. At least as supplement, I would recommend to compare Nivo+Ipi versus PD1 mono subpopulations.

2) Since pretreatment before ICI was an important factor: what type were these pretreatments? Targeted therapies?

3) For reinduction, why was only 1/21 patients treated with Nivo+Ipi?

4) Figure 4: instead of "Temodal", "temozolamide" should be written. 

Author Response

A Point-by-point responses the reviewer 2 comments:

Point 1: I am surprised that there are no differences with regard to Nivo+Ipi versus PD1 monotherapy as primary ICI. At least as supplement, I would recommend to compare Nivo+Ipi versus PD1 mono subpopulations.

Response 1: Table 1 specifies the patients' demographics and their treatment characteristics. Inclusion of the patients in this study required at least non-progression on 6 months of treatment and treatment discontinuation (>96% of the included patients achieved a PR or a CR). Thus, the chances for response or for becoming treatment-free, which are probably very different between Ipi+Nivo and Anti-PD-1 monotherapy, cannot be compared here and are not intended to be evaluated in this sort of analysis. We compared the patient characteristics between patients who progressed and those who did not progress after treatment discontinuation. The results show that patients from the Ipi-Nivo group had a similar odds ratio for disease progression after treatment discontinuation, to the patients from the anti PD-1 monotherapy (p=0.169). This suggests that upon achieving a response followed by treatment cessation, the chances for disease progression are not dependent on the immune checkpoint modality. See Table 1, Treatment characteristics.

Point 2: Since pretreatment before ICI was an important factor: what type were these pretreatments? Targeted therapies?

Response 2: This is an important point and we thank the reviewer for adding it. We added in Table 1 the previous treatments given, lines 114-116. Those were: ipilimumab (n=15), targeted therapy (n=6), pembrolizumab (n=3), ipilimumab and nivolumab (n=2). 

Point 3: For reinduction, why was only 1/21 patients treated with Nivo+Ipi?

Response 3: This was a retrospective study and reasons for choosing one therapy over the other were not collected. One should also consider that the recently published data on re-induction was not known at the time. Furthermore, ipilimumab and nivolumab as a second line is currently not reimbursed in Israel. Nonetheless, in some cases the physicians could recall their decisions: in 7 patients radiotherapy or surgery were combined so toxicity was of concern; in 3 patients the disease burden was low and monotherapy seemed enough at the time; Three patients had poor PS and 2 were octogenarians upon recurrence, so again toxicity was of concerns here. 

Point 4: Figure 4: instead of "Temodal", "temozolamide" should be written. 

Response 4: Corrected. see Figure 4.

Round 2

Reviewer 2 Report

none